# Mixed-methods study in England and Northern Ireland to understand young men who have sex with men's knowledge and attitudes towards human papillomavirus vaccination

Joanna May Kesten,[1] Carrie Flannagan,[2] Eimear Ruane-McAteer,[3] Samuel William David Merriel,[4] Tom Nadarzynski,[5] Gilla Shapiro,[6] Zeev Rosberger,[6] Gillian Prue[3]

For numbered affiliations see end of article.

**Correspondence to**
Dr Joanna May Kesten;
jo.kesten@bristol.ac.uk

## ABSTRACT

**Objectives** Men who have sex with men (MSM) are at greater risk for human papillomavirus (HPV)-associated cancers. Since 2016, MSM have been offered the HPV vaccination, which is most effective when received prior to sexual debut, at genitourinary medicine clinics in the UK. In September 2019, the national HPV vaccination programme will be extended to boys. This study aimed to understand young MSM's (YMSM) knowledge and attitudes towards HPV vaccination.

**Design** Questionnaires assessed YMSM demographics, sexual behaviour, culture, knowledge and attitudes towards HPV vaccination and stage of vaccine decision-making using the precaution adoption process model. Focus groups explored sexual health information sources, attitudes, barriers and facilitators to vaccination and strategies to support vaccination uptake. Questionnaire data were analysed using descriptive statistics and focus group data were analysed thematically.

**Setting** Questionnaires were completed online or on paper. Focus groups were conducted within Lesbian Gay Bisexual Transgender Queer organisational settings and a university student's union in England and Northern Ireland.

**Participants** Seventeen YMSM (M=20.5 years) participated in four focus groups and 51 (M=21.1 years) completed questionnaires.

**Results** Over half of YMSM were aware of HPV (54.9%), yet few (21.6%) had previously discussed vaccination with a healthcare professional (HCP). Thematic analyses found YMSM were willing to receive the HPV vaccine. Vaccination programmes requiring YMSM to request the vaccine, particularly prior to sexual orientation disclosure to family and friends, were viewed as unfeasible. Educational campaigns explaining vaccine benefits were indicated as a way to encourage uptake.

**Conclusions** This study suggests that to effectively implement HPV vaccination for YMSM, this population requires clearer information and greater discussion with their HCP. In support of the decision made by the Joint Committee on Vaccination and Immunisation, universal vaccination is the most feasible and equitable option. However, the absence of a catch-up programme

## Strengths and limitations of this study

► This is the first study in the UK to explore young men who have sex with men's (YMSM) knowledge and attitudes towards human papillomavirus (HPV) vaccination.

► Use of a theoretical model of behavioural change facilitates clear conceptualisation of health behavioural change and YMSM's stage of HPV vaccine decision-making.

► The qualitative component obtained a diverse range of views of YMSM in England and Northern Ireland.

► Survey findings should be interpreted with caution due to the sample size.

will leave a significant number of YMSM at risk of HPV infection.

## INTRODUCTION

Human papillomavirus (HPV), the most common sexually transmitted infection (STI) worldwide,[1] has serious health consequences for men and women. HPV is recognised as a causative agent in cervical cancer, and is associated with anogenital tumours, oropharyngeal cancers and genital warts.[2] While boys and girls aged 12–13 years are vaccinated in school in Australia,[3 4] the current UK strategy of vaccinating all girls aged 12–13 years does not protect young men who have sex with men (YMSM) against HPV infection and related diseases[5] as they will not benefit from herd immunity.[6]

A Joint Committee on Vaccination and Immunisation's (JCVI) statement on MSM HPV vaccination[7] in 2015 recommended that vaccination programmes be extended to MSM aged up to 45 years via genitourinary

medicine (GUM) clinics. Mathematical modelling suggested that for MSM aged 40 or over, HPV vaccination in GUM clinics was likely to be an effective and cost-effective method of reducing HPV-related disease burden in MSM in England[8] and elsewhere. Northern Ireland, Scotland and Wales are currently offering the HPV vaccine to MSM attending GUM clinics. Following a pilot programme in England[9] which found suboptimal uptake (45%) and did not report completion rates, vaccination is now offered in GUM clinics. Hence, it is important to assess the reasons why MSM might not be willing to accept the vaccine through targeted HPV vaccination. An interim statement in July 2017 suggested that given the current high uptake in females, extending immunisation to all adolescent males is 'highly unlikely to be cost-effective in the UK' (p13).[10 11] In July 2018, the JCVI recommended that the national HPV vaccination programme should be extended to include adolescent boys. It is planned that the programme, beginning in September 2019, will include boys aged 12/13 (England, school year 8; Northern Ireland, school year 9). Although some may now query the importance of the MSM programme (particularly for YMSM), this will still be valid for a number of years because the government have indicated that they will not initiate a catch-up programme for boys so there are still a significant number who will remain unprotected. Indeed, it is worth noting that it took 5 years of deliberation by the JCVI to make this decision and that boys aged 13 plus will not be offered the vaccine in schools.

The absence of a catch-up vaccination programme leaves many UK YMSM without funded access to the HPV vaccine before exposure to HPV.[12] There is often a delay between the age of first sexual contact with another man and disclosure of sexual orientation to a healthcare professional (HCP),[13] as a result, it is likely that MSM will have multiple sexual partners before attending a GUM clinic resulting in increased risk of HPV acquisition.[14]

A systematic review found that MSM HPV vaccine knowledge was low and MSM did not consider themselves at risk of infection, although over half would accept the vaccine if they were offered it.[12] Most of these studies were conducted in North America (and none in the UK), with MSM over 26 years of age. Minimal attention has been given to the knowledge and attitudes towards HPV vaccination among adolescent and YMSM (aged 16–24 years). This is an important area for research because MSM may acquire HPV at a young age, close to their sexual debut (the age of which is decreasing).[14] This study aimed to examine the knowledge and attitudes of UK YMSM towards HPV vaccination to inform policy and practice recommendations for accessing this hard to reach group, supporting vaccination uptake and the optimisation of protection from HPV. Despite the changes to the vaccination programme made since this research was conducted in 2017, in the absence of a catch-up programme, the newly implemented universal programme will cover not all YMSM. Therefore, understanding YMSM knowledge and attitudes to HPV remains relevant in the UK. Our

findings are also relevant for guiding other programmes internationally that do not have a gender-neutral programme and are considering implementation of a programme for YMSM.

## METHODS
### Study design
We conducted questionnaires and focus groups with YMSM aged 16–24 years. The two substudies are described separately below.

### Questionnaire study
#### Data collection
The survey was administered online using Survey Monkey and on paper and advertised via various Lesbian Gay Bisexual Transgender Queer (LGBTQ) organisations on social media (Twitter and Facebook). We have combined both the online and pen and paper completions for this paper.

#### Measures
The questionnaires (online supplementary material A) assessed demographics (adapted from Hickson et al)[15]; sexual behaviour (adapted from Sadlier et al)[16]; culture (adapted from Zou et al)[17] and HPV vaccine stage of decision-making using the precaution adoption process model (PAPM).[18] The PAPM has six stages of behavioural change decision-making and has been used to examine knowledge and attitudes to HPV vaccination.[19] Those who indicated awareness of the HPV vaccine were asked to complete validated HPV knowledge/attitudes scales.[20 21]

#### Patient and public involvement
The HPV knowledge/attitude questionnaire scales were adapted for use with MSM through consultation with an expert panel including a key stakeholder group (The Rainbow Project (TRP)) and MSM focus groups.

YMSM were not involved in the development of the qualitative component of this study, however, staff from TRP helped develop the study design and documentation.

The findings will be disseminated to YMSM via social media and TRP.

### Focus group study
#### Data collection
We aimed to achieve data saturation[22] by recruiting 8–10 YMSM per focus group with a mix of social background, age, ethnicity and religion. YMSM were defined through self-identification as male (including transgender male), at or over the age of sexual consent, sexually attracted to men or had sex with a man.[14] Age inclusion criteria were based on WHO's definition of 'young': 15–24 years. A minimum of 16 years was specified as it is the age of sexual consent in the UK.

For the focus groups, potential participants were provided with written study information, and asked to register their interest at local LGBTQ organisations, university information days, university student union

clubs and societies, and secondary school LGBTQ groups. Organisations advertised the study through social media and snowball sampling was employed.

CF conducted the focus groups within LGBTQ organisational settings and a university student's union building.

Prior to the focus groups, participants were asked to complete the questionnaire (described in the Questionnaire study section).

The focus group topic guide (online supplementary material B) was applied flexibly to allow for emergent issues and began by exploring sources of sexual health information and advice before engaging in sexual activity (not presented here). Perceptions of HPV risk in relation to six other STI's were then discussed using a sorting task in which a list of STIs were ordered by what is least to most concerning (findings not reported here). Attitudes towards HPV vaccine, barriers and facilitators to vaccination and possible intervention strategies to support vaccination uptake were explored. Experiences of disclosing sexual orientation to HCP were also discussed. All participants were informed that the HPV vaccine was most protective if received prior to first sexual encounter. Participants were asked to reflect as to how they would have viewed taking the vaccine when they were 12–13 years.

### Analysis
#### Questionnaire study
Questionnaire data were inputted to SPSS V.12 and analysed descriptively with frequencies and proportions reported for categorical data and mean and standard deviation for continuous data. Due to a lack of statistical power, it was not possible to use inferential statistics for analysis. Participants' PAPM vaccine decision-making stage was classified into six stages: unaware, unengaged, undecided, decided not to vaccinate, decided to vaccinate and those who had already been vaccinated.[19] If participants indicated they were not sexually active they were asked to skip the sexual contact questions. If they indicated that they had never heard of the HPV vaccine they did not complete the knowledge/attitude scores. Knowledge and attitudes held by participants about HPV and HPV vaccination were analysed using descriptive statistics.

#### Focus group study
Focus groups were audio recorded, transcribed verbatim, anonymised and analysed thematically[23] using QSR NVivo (V.10.0). This approach was chosen because it offers a clear analysis process while remaining flexible.[23] JMK and CF independently coded the first transcript systematically, line-by-line, compared their coding and reached consensus on the definition of codes. These initial codes, which captured features of interest in the data, were then applied to the remaining transcripts. The content of all the codes was read and compared with each other to iteratively refine and cluster codes into themes and subthemes. For example, duplicate codes with synonymous meanings were collapsed. A description of each theme capturing

instances of divergence was then written by JMK. At each stage, findings were verified and discussed by the research team to assess the accuracy and credibility of the interpretation, promote inter-rater reliability and ensure rigour.

Participants were not provided any financial remuneration for their time.

## RESULTS
### Participant characteristics
Between September 2016 and March 2018, questionnaires were completed by 51 YMSM. From this 51, four focus groups in Northern Ireland (n=3) and England (n=1) were conducted between September and December 2016 with 17 YMSM who had completed the questionnaires (table 1). Focus group size ranged from two to six participants and lasted approximately 45 min.

### Questionnaire results
The majority (n=49) were sexually active and reported both oral and anal intercourse in the past 12 months (n=35), a wide range of partner numbers (M=5 partners, range 0–25), and 'sometimes' (n=17) or 'never' (n=16) used condoms. Twenty-nine (57%) participants had accessed sexual health services (table 2).

Nineteen participants (37%) had never heard of HPV and did not complete the rest of the questionnaire. Of those who had heard of HPV in accordance with the PAPM, 18% were in the 'decided to act' stage of vaccine decision-making (stage 5), none had decided that they did not want the vaccine (stage 4) and 22% had already been vaccinated (stage 6) (table 3).

Of those who were aware of HPV (n=28), knowledge of HPV and the HPV vaccine was generally high; mean items correct 65% (M=13.3, SD 4.7) and 60% (M=3.3, SD, 1.2), respectively. However, there was a wide variation in knowledge scores (HPV range, 3–20; HPV vaccine range, 0–5) (table 3). Participants were aware that HPV affected men, the method of HPV transmission, and that vaccination was most effective if given prior to sexual debut. However, awareness of the link between HPV and genital warts and the severity of an HPV infection was lower as the majority of YMSM thought HPV infection always required treatment and that infection with HPV would always lead to health problems (table 3).

Thirty-three participants (65%) reported that HPV vaccination had never been discussed with or recommended by an HCP (table 3). The mean age participants were willing to disclose their sexuality to an HCP was 18.3 years (range=11–23, SD=2.40) (table 3). The most comfortable setting cited to receive the HPV vaccine was primary care or LGBTQ-specific services, rather than GUM clinics (table 3).

### Qualitative results
Two main themes and several subthemes were elicited from the thematic analysis: (1) Willingness to be vaccinated and (2) Implementation recommendations.

**Table 1** Participant characteristics

| Participant characteristics | Questionnaire participants | | | Focus group participants (subset of questionnaire participants) | | |
|---|---|---|---|---|---|---|
| | M (SD) | Range | N | M (SD) | Range | N (% of sample) |
| Age (years) | 21.06 (2.6) | 16–24 | | 20.5 (2.73) | 16–24 | 18* (100) |
| Ethnicity | | | | | | |
| White | | | 44 (86.3) | | | 15 (83.3) |
| Other | | | 6 (11.8) | | | 3 (16.6) |
| Missing | | | 1 (1.9) | | | |
| Location | | | | | | |
| Northern Ireland | | | 36 | | | 13 (72.2) |
| England | | | 15 | | | 5 (27.8) |
| Education | | | | | | |
| Full-time education | | | 26 | | | 11 |
| Employed full time | | | 17 | | | 4 |
| Employed part time | | | 5 | | | 1 |
| Unemployed | | | 2 | | | 1 |
| Missing | | | 1 | | | 1 |
| Group size | | | | | | |
| Focus group 1* | | | | | | 6 |
| Focus group 2 | | | | | | 2 |
| Focus group 3 | | | | | | 4 |
| Focus group 4 | | | | | | 5 |

*One participant completed the questionnaire and left before the focus group began due to time constraints.

Anonymous quotes illustrating the key themes are presented below. Minimal differences in attitudes towards HPV between geographical settings were found.

### Willingness to be vaccinated
Despite a perceived lack of knowledge about HPV and the vaccine and the threat posed to men, most participants were willing to receive the vaccine and wanted more information.

> P1: I only knew about it because of the cervical cancer (…)
> P2: I didn't even know that was what it was for.
> P1: I didn't know even if like that would apply to us, so I don't even know what the dangers are.
> Focus group 2

Participants were motivated to receive the vaccine to protect their health and a small number of participants suggested that the cost and number of doses of the vaccine were not barriers to vaccination.

> I'm not going to say like get rid of worry because you still have to…it's your sexual health, but it would be safer in a sense (…) I'm better protected – I think would be a better way of putting it. So, I think my own health would encourage me more [to ask or accept the HPV vaccine]. I'd rather be protected than not protected.
> Focus group 3, unidentifiable speaker

### Implementation recommendations
Across the focus groups, recommendations to support the implementation of the HPV vaccine were gathered and grouped into two subthemes: 'promoting and raising awareness of the vaccine' and 'identifying and offering YMSM the HPV vaccination'.

#### Promoting and raising awareness of the vaccine
Better understanding of the benefits and side effects of the vaccine were expected to encourage uptake. To promote the vaccine and inform YMSM, awareness campaigns and advertisements on the internet, radio, television, social media, in University society's, LGBTQ youth groups and dating apps were suggested.

> For this generation particularly, social media and TV ads and newspapers – well, probably not newspapers, but radio ads as well. You know, a campaign around getting people vaccinated, I think that would be very beneficial for young people these days.
> Focus group 3, unidentifiable speaker

**Table 2** Sexual contact and relationships

| Sexual contact and relationships | M (SD) | Range | N (%) |
|---|---|---|---|
| Have you ever in the past had sex with a man or do you plan to in the future? | | | |
| Yes | | | 49 (96.08) |
| No | | | 1 (1.96) |
| Missing | | | 1 (1.96) |
| Relationship status | | | |
| Single | | | 26 (50.98) |
| In a relationship | | | 21 (41.18) |
| Cohabiting | | | 2 (3.92) |
| Civil partnership | | | 1 (1.96) |
| Missing | | | 1 (1.96) |
| Are you sexually active? | | | |
| Yes | | | 38 (74.51) |
| No | | | 8 (15.69) |
| Missing | | | 5 (9.8) |
| How many male sexual partners have you had in the past 12 months? | 5 (6) | 0–25 | |
| What type of intercourse have you had in the past 12 months? | | | |
| Oral only | | | 3 (5.88) |
| Anal only | | | 2 (3.92) |
| Both oral and anal | | | 35 (68.63) |
| Neither | | | 3 (5.88) |
| Missing | | | 8 (15.69) |
| In the past 12 months have you used condoms? | | | |
| Always | | | 9 (17.65) |
| Sometimes | | | 17 (33.33) |
| Never | | | 16 (31.37) |
| Prefer not to say | | | 1 (1.96) |
| Missing | | | 8 (15.69) |
| Do you access sexual health services? | | | |
| Yes | | | 29 (56.86) |
| No | | | 14 (27.45) |
| Missing | | | 8 (15.69) |

Participants suggested including information about the vaccine for YMSM in primary care and the sexual health education curriculum in schools. Indeed, it was noted that there is a lack of MSM-specific sexual health and relationship information provided in the latter.

When you're receiving that [heterosexual relationship education] in school, (…) it just reinforces the fact that you're (…) not relating to it means that you're not normal like everyone else, so you don't want to speak about it. So it would just be better if it [HPV vaccine education for MSM] was just part of that education anyway.

Focus Group 2, participant 1

**Table 3** HPV vaccine: culture, awareness and stage of decision-making

| | M (SD) | Range | N |
|---|---|---|---|
| GP aware of sexuality | | | |
| Yes | | | 22 (43.14%) |
| No | | | 17 (33.33%) |
| Not sure | | | 8 (15.69%) |
| Missing | | | 4 (7.84%) |
| Willing to disclose MSM status to HCP to receive HPV vaccine? | | | |
| Yes | | | 41 (80.39%) |
| No | | | 3 (5.88%) |
| Not sure | | | 3 (5.88%) |
| Missing | | | 4 (7.84%) |
| If yes, at what age? | 18.3 (2.4) | 11–23 | |
| Has an HCP ever recommended an HPV vaccine to you? | | | |
| Yes | | | 11 (21.57%) |
| No | | | 33 (64.71%) |
| Not sure | | | 1 (1.96%) |
| Missing | | | 6 (11.76%) |
| Discussed HPV vaccination with HCP | | | |
| Yes | | | 10 (19.61%) |
| No | | | 34 (66.67%) |
| Missing | | | 7 (13.73%) |
| Most comfortable setting to receive HPV vaccine (some ticked more than one option) | | | |
| Genitourinary medicine | | | 17 (33.33%) |
| Primary care | | | 30 (58.82%) |
| Lesbian gay bisexual | | | 33 (64.71%) |
| Transgender organisations | | | 1 (1.96%) |
| Non-LGBTQ-specific sexual health provider | | | 2 (3.92%) |
| HIV clinic | | | 1 (1.96%) |
| Prior awareness of HPV | | | |
| Yes | | | 28 (54.9%) |
| No | | | 19 (37.25%) |
| Missing | | | 4 (7.84%) |
| PAPM (stage of vaccine decision-making) | | | |
| Stage 2 unengaged: I have never thought about vaccination against HPV | | | 17 (33.33%) |
| Stage 3 undecided: I am undecided about vaccination against HPV | | | 2 (3.92%) |
| Stage 4 decided not to act: I have decided and do not want to vaccinate myself against HPV | | | 0 |

Continued

**Table 3** Continued

| | M (SD) | Range | N |
|---|---|---|---|
| Stage 5 decided to act: I have decided and I do want to vaccinate | | | 9 (17.65%) |
| myself against HPV | | | |
| Stage 6 acted: I have already been vaccinated against HPV | | | 11 (21.57%) |
| Missing | | | 12 (23.53%) |
| Knowledge scores | | | |
| HPV knowledge score (max 20) | 13.3 (4.7) | 3–20 | 27 |
| HPV vaccination knowledge score (max 5) | 3.3 (1.2) | 0–5 | 27 |

GP, general practitioner; HCP, healthcare professional; HPV, human papillomavirus; LGBTQ, Lesbian Gay Bisexual Trans Queer; MSM, men who have sex with men; PAPM, precaution adoption process model

### *Identifying and offering YMSM the HPV vaccination*

The ideal pre-exposure timing for vaccination and the fluid, undefined nature of sexual preferences at a young age were perceived as barriers to identifying eligible recipients. There were mixed feelings about whether it would be acceptable for HCPs to ask boys (<16 years) to disclose their sexuality for this purpose due to concern about parents being informed and a lack of a trusting relationship. It was, however, also noted that questions about sexuality need to be normalised, particularly in primary care.

Interviewer: If everybody was getting the HPV vaccine…

Participant 2: That's probably what they should do, because, I mean, (…) someone might think now, oh, I'll never have sex with a man, but then, later down the line, they might do.

Focus group 4

The focus group participants wanted the benefits of vaccination to be explained and for the vaccine to be offered in a natural, relaxed manner, opportunistically, rather than having to request it. Participants felt that they would be unlikely to request the vaccine because they would need more knowledge and they felt too uncomfortable.

Participant 2: As long as there was someone professional telling me what's it about, what's it going to do, and what it could do if it goes wrong.

Focus group 2

Participants reckoned it was not feasible to expect young boys to identify themselves for the HPV vaccine when they potentially had not disclosed or decided their sexual orientation. There was also a preference for not singling boys out by their sexuality when offering the vaccine. Similarly, receiving the vaccine confidentially was important because the potential for bullying and embarrassment would act as barriers. It was noted by participants that universal vaccination of all boys would avoid these problems. A young person seeking sexual health advice represented an opportunity to identify eligible boys. However, this is likely to occur post sexual encounter—after the risk of exposure to the virus.

I would want them to approach me. I wouldn't go out of my way to go and get it.

Focus group 3, unidentifiable speaker

Interviewer: So then you're asking Year 8 and 9 that age group (…) -

Unidentifiable participant: To basically out themselves…

[Agreement] Interviewer: Do you see that as being a feasible scenario?

Unidentifiable participant: No.

Unidentifiable participant: Absolutely not.

Unidentifiable participant: The only kind of way round that is if every like male child is also vaccinated, but (…) obviously they won't do that because in terms of cost of vaccines.

Focus group 3, unidentifiable speaker

Participant 2: When you get your vaccinations in school, you all, (…) used to go in to get your vaccinations [as a class]. If it were like that, I wouldn't do it, because I wouldn't like anyone seeing.

Focus group 4

Participant 6: Why wouldn't it be offered to like young males in school, (…) so it was like before like presumably anybody had had sex (…). A lot more people would get it that way.

Focus group 1

There were mixed feelings about general practitioners (GPs) or specialist sexual HCPs offering the vaccine. The relationship with the HCP was important; if YMSM have a good relationship with their GP then being offered the HPV vaccine by them is preferable. In contrast, a small number would feel more comfortable being offered the vaccine by someone they trust from a community LGBTQ group or local sexual health centre. A comment was also made about the nature of the vaccine being related to sexual health meaning it made more sense/was easier to offer it via specialist services. However, prior to disclosure or sexual activity, the participants commented that boys may not engage with or know about sexual health or LGBTQ organisations so offering the vaccine in these settings may represent a barrier.

Telling your family GP you're gay before you've told your family would be a big no I think because the GP might go back and tell your parents and then out you.

Focus group 3, unidentifiable speaker

If you have to go and ask about it and ask for it, who would you ask because you wouldn't be able to come here [Community LGBTQ group] because you wouldn't know here existed.

Focus group 4

Written invitations from GPs offering the vaccine to eligible patients were also suggested. However, this would require boys to identify as MSM when registering or being asked about their sexuality by an HCP. A small number of accounts suggested that it would be acceptable to refer patients to receive the HPV vaccine in sexual health clinics if it was not available in a GP setting. Offering the vaccine in schools when YMSM are beginning to have their first sexual encounters was suggested. Similarly, the school nurse was a trusted individual for some and therefore may be an acceptable person to provide the vaccine.

## DISCUSSION

This is the first study to examine the views of YMSM towards the HPV vaccine in the UK. Despite being sexually active and willing to disclose sexual orientation to receive the vaccine, most participants had never been recommended the HPV vaccine, suggesting that MSM are not being offered the vaccine at the most opportune time. The data also suggested that HPV knowledge in YMSM is low, with almost half of participants being unaware of HPV or the vaccine. YMSM were willing to receive the vaccine but wanted additional information about HPV and the vaccine. Given the reluctance to disclose information about sexuality to HCPs (prior to disclosure to significant others), the wide range of sexual partner numbers, and lack of consistent contraceptive use, combined with the importance of supporting vaccination prior to potential exposure, the findings highlight significant barriers to MSM accessing the vaccine. Early provision of information was recommended through awareness campaigns, advertisements and the school health education curriculum. However, even with enhanced awareness, programmes that rely on YMSM to present for vaccination (particularly prior to sexual orientation disclosure) were not viewed as feasible. Furthermore, preferences for GPs or specialist HCPs offering the vaccine were dependent on the relationship with the HCP. Offering the vaccine to MSM in schools was thought to be acceptable. We accept that many of these issues will now hopefully be addressed by the extension of the current female vaccination programme to boys in September 2019, although the lack of catch-up programme for boys would indicate that there is still a need for the vaccine programme to target YMSM for at least the next 6 years as a significant number of YMSM will be a risk of HPV infection. In addition, these findings offer insights into barriers to vaccination for YMSM which will be useful if the uptake of a universal vaccination programme is low.

### Strengths and limitations

This is the first study in the UK exploring this topic with YMSM. By conducting this research in more than one setting, we can comment on the transferability of our findings; we found minimal differences in attitudes towards HPV between settings. The use of a theoretical model of behavioural change, the PAPM, also facilitates clear conceptualisation of health behavioural change and YMSM's stage of HPV vaccine decision-making.

We aimed to continue data collection until saturation, however, recruitment difficulties and the study time frame meant that the decision to cease recruitment was pragmatic. The sensitivity of the topic, the hard to reach population and the lack of monetary compensation for the participant's time are possible explanations for this. Therefore, the findings must be read with caution. Those who self-selected to participate may be more comfortable with their sexuality than those who did not agree. Indeed, recruiting through LGBTQ organisations narrowed our participant pool to those engaged with these services who had disclosed their sexual orientation. The small sample size for the quantitative data resulted in a lack of statistical power to analyse data using inferential statistics and should be considered in generalising beyond the study sample. Small sample size in research with sexual and gender minorities is a recognised limitation.[24] The interview sample age range of 16–24 years is older than the target population for the vaccine—12–13 years. Although the participants were asked to consider how they would view the vaccine and strategies to implement it among YMSM, it is unclear whether current YMSM share similar attitudes.

### Implications for research and practice

The reluctance of YMSM to discuss their sexuality with HCPs before they have disclosed to significant others has important implications for the success of an HPV vaccination programme. Previous research shows that MSM disclosing their sexuality to significant others, visiting HCPs in the past year and previous STI diagnosis predict disclosure to an HCP.[25] In the absence of a catch-up programme for boys, additional measures to support YMSM to access the vaccine are necessary. For instance, information may need to be provided to young men outside of healthcare settings including educational contexts during sex and relationship education or HCPs may need to take an active role in opportunistically providing information during consultations for non-sexual health related matters. To support the latter, GPs and other HCPs may require additional education and training.[26 27]

### Comparison to existing literature

A lack of knowledge does not appear to deter MSM willingness to be vaccinated.[28] However, MSM in this study and in others[28] desired more information. Poor knowledge of the HPV vaccine among YMSM has also been reported previously.[12 29]

Other qualitative work with MSM has shown support for vaccinating all adolescent boys in school in part to protect against stigma arising from vaccination policies targeting MSM.[30] This would also remove the barrier of MSM having to request the vaccination, especially prior to sexual debut.[27]

Our finding that MSM are unlikely to disclose sexual orientation to a HCP prior to sexual debut, has been reported elsewhere,[13] suggesting that HPV vaccine programmes delivered by HCPs would be of 'limited benefit'.[13] Participants in our study recommended that the vaccine is offered by HCPs rather than expecting them to request it; however it is unclear whether initial reluctance to disclose sexuality would prevent vaccination uptake. The absence of an HCP's recommendation has previously been identified as a barrier to vaccination.[31] A new National Health Service England standard recommending 'sexual orientation monitoring' whereby patients aged 16 and over are asked to disclose their sexual orientation at every face-to-face appointment may help to identify those eligible for vaccination.[32] Although this standard would not help identify those younger than 16 years who may benefit from the vaccine.

Previous research has found that most MSM have positive attitudes towards vaccinations against STIs and would be willing to receive the HPV vaccine.[29 30] However, individual and systemic barriers such as access to sexual health clinics, disclosure of sexual orientation, concern about confidentiality or belief that HPV vaccine is not effective after sexual debut, may compromise the effectiveness of vaccination strategies.[30] Additionally, perceptions that HPV is relatively uncommon and harmless may lead to low desirability of the vaccine resulting in suboptimal coverage and therefore reduced cost-effectiveness.[30]

In line with our findings, awareness-raising strategies are vital to HPV vaccination programme success.[29 33–35] To raise awareness and motivate vaccine uptake, a public health campaign may be necessary.[28] When developing strategies for HPV vaccination programmes, stakeholders can learn from the introduction of vaccinations such as hepatitis B and should engage with the target population and coordinate between stakeholders to ensure consistent messages.[33] In addition, offering the HPV vaccination to MSM alongside other vaccinations and during STI screening consultations has been recommended.[29 35]

## CONCLUSIONS

This study suggests that UK YMSM's are willing to receive the HPV vaccine. However, the UK's current HPV vaccine programme that relies on MSM to present for vaccination (particularly prior to sexual orientation disclosure) was not viewed as feasible. The importance of supporting vaccination prior to potential virus exposure combined with the reluctance to disclose information about sexual orientation means personal knowledge and awareness of the HPV vaccine is important, therefore, early provision of information is recommended. Offering the vaccine

in healthcare and education settings may be acceptable, although the barriers to this channel of provision may mean that, in support of the decision made by the JCVI, universal vaccination is the most feasible and equitable option. However, in the absence of a catch-up programme, there is still a need for the UK vaccine programme to target YMSM as a significant number will remain at risk of HPV infection. These findings also help guide other programmes internationally that do not have a gender-neutral programme and are considering implementation of a programme for YMSM.

**Author affiliations**
[1]National Institute for Health Research (NIHR) Health Protection Research Unit (HPRU) in Evaluation of Interventions and NIHR Collaboration for Leadership in Applied Health Research and Care (CLAHRC) West, University of Bristol, Population Health Sciences, Bristol Medical School, Bristol, UK
[2]Institute of Nursing and Health Research, School of Nursing, Ulster University, Newtownabbey, UK
[3]School of Nursing and Midwifery, Medical Biology Centre, Queen's University Belfast, Belfast, UK
[4]Centre for Academic Primary Care, University of Bristol, Bristol, UK
[5]School of Social Sciences, University of Westminster, London, UK
[6]Department of Psychology, McGill University, Montreal, Canada

**Acknowledgements** We would like to thank the MSM and TRP stakeholders who informed the development of the HPV knowledge/attitude questionnaire scales.

**Contributors** JMK, GP and ER-M drafted the manuscript and JMK led the analysis of the qualitative data supported by CF. CF conducted the focus groups and questionnaires. GP, JMK and SWDM conceived the research question. ZR's research team developed the original questionnaires and use of the PAPM in college males and parents of young children eligible for the HPV vaccine (prior to adaption for MSM). GP led the research team. ER-M and GP conducted the analysis of the questionnaire data and led the writing of these sections of the paper. All coauthors (JMK, CF, ER-M, SWDM, TN, GS, ZR and GP) developed the research question, methodology and supported the management of the project. All authors (JMK, CF, ER-M, SWDM, TN, GS, ZR abd GP) have read, contributed to and approved the final manuscript.

**Funding** This study was funded by an Innovation award (#22091) from Cancer Research UK/BUPA Foundation. JMK is partly funded by National Institute for Health Research (NIHR) Collaboration for Leadership in Applied Health Research and Care West (CLAHRC West) at University Hospitals Bristol NHS Foundation Trust and NIHR Health Protection Research Unit in Evaluation of Interventions. The views expressed in this article are those of the author(s) and not necessarily those of the NIHR, or the Department of Health and Social Care. SWDM was the recipient of an academic clinical fellowship from NIHR. GS was supported by the Vanier Canada Graduate Scholarship and Queen Elizabeth II Diamond Jubilee Scholarship programmes.

**Competing interests** None declared.

**Patient consent for publication** Not required.

**Ethics approval** This study was approved by the Queens University Belfast, School of Nursing and Midwifery Research Ethics Committee (39.GPrue.05.16.M8.V2). Written informed consent from each participant was obtained prior to participation in the focus groups.

**Provenance and peer review** Not commissioned; externally peer reviewed.

**Data sharing statement** No additional data are available.

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
