## [Reviewer comments · BMJ Open]

ARTICLE DETAILS

TITLE (PROVISIONAL)	A mixed-methods study in England and Northern Ireland to understand young men-who-have-sex-with-men's knowledge and attitude towards human papillomavirus vaccination
AUTHORS	Kesten, Joanna; Flannagan, Carrie; Ruane-McAteer, Eimear; Merriel, Samuel; Nadarzynski, Tom; Shapiro, Gilla; Rosberger, Zeev; Prue, G

VERSION 1 - REVIEW

REVIEWER	Holly Fontenot Boston College, USA
REVIEW RETURNED	07-Aug-2018

GENERAL COMMENTS	Review: Overall, very important topic and timely for the UK to consider for policy change to be more inclusive. Below are suggested revisions to add to the quality of the paper. Methods: Pg 6 line 52- online via what platform, qualtrics?, secure? And how were advertisements done? Posters, ads on the organizations facebook pages or websites? Please describe how you did ads for the quant and qual parts of the study Page 7 line 5- race and ethnicity. Also... this data was not presented in any table or text? In methods ensure consistency in language for where you are recruiting. You say LGBTQ organizations and then you say advocacy groups. I would keep with just organizations. LGBT is defined on page 7 but you first use the abbreviation earlier in the section on page 6. You describe how you recruited for the focus groups, but you did not describe the ads for how you recruited for the online survey. Data Collection: Page 8, line 3-11. Did you also discuss effectiveness and recommendation to vaccinate even if the participant has past sexual experience, "catch up vaccination"- did they voice desire to go get vaccinated now? I suggest a subheading to describe measures, then a separate heading to discuss data collection Authors never really explained how quant data was recruited for or collected or where data was downloaded, managed, stored for analysis.
---

	Where the focus groups first then they informed the quant survey, did the focus groups and the public help to inform the survey? Make that clear. The section about the public is very confusing and not needed if that information is incorporated into the measures section of the paper. Analysis: Break into 2 sections, 1) quant, 2) qual. For the quantitative it was all descriptive analyses, what analyses did you use, what software did you use? Page 8 line 50. Please describe how you defined codes, then used that framework to move the subsequent coding forward, codebook? So beyond just comparing the 1st level codes, you build consensus, then what was the plan, you must have agreed on definitions for codes to continue to move forward. Page 9, line 10. Any remuneration for the participants? Survey or focus groups- spell that out here so when you mention that in the discussion it makes sense. Results: You can just say approximately 45 minutes, no need to give mean or range for how long the focus groups lasted. Page 11. Tell me the 2 themes in line 6 in a sentence and then go into them in the subheadings Page 12. Can you give a sentence here again laying out the sub-themes before you go into them. More richer description of participant discussion would be helpful, then use the quotes of exemplars for the findings. Discussion: Consider viewing these 2 articles: (1) Apaydin, Vaccine, 2018, DOI: 10.1016/j.vaccine2018.06.057 and (2) Fontenot, Vaccine, 2016, 34, pg 6209-6216 They way in which a HCP recommends HPV vaccine to sexual and gender minority youth (affirming care) is important for vaccine uptake. Page 17. Line 26. “the sample size is however, considered appropriate as it has obtained a diverse range of views of YMSM”—how? What is the race, ethnicity- urban/ rural, educated, uneducated, homeless? Might be an overstatement Strengths and limitations could be re-worked. End of that paragraph... does the UK have a vaccine “catch up” recommendation for those older than 12? Tables: Consider a table with sample characteristics for the entire group with break outs for quant and qual groupings. Give general demographics and add race/ethnicity.
--	--

REVIEWER	Jo Waller UCL, UK
REVIEW RETURNED	24-Jan-2019

GENERAL COMMENTS	I'm afraid the relevance of this paper has decreased significantly since it was submitted in June 2018, following the recommendation by the JCVI that gender neutral HPV vaccination be introduced in the UK. I'm not sure why there's been such a long delay with reviewing the paper, which I only received in January 2019, and I sympathise with the inevitable frustration that the authors must feel about this.
--

If the authors are still keen to publish the paper, major edits will have to be made to the Introduction and Discussion to re-frame the paper in the context of HPV vaccination for boys being introduced from the next academic year. This may be possible.

Overall, the study addresses what was an important issue of how to maximise engagement with HPV vaccination among MSM. Much of the discussion (e.g. about trying to ascertain sexual orientation at a very young age, before initiation of sexual activity) is now moot, but there may still be findings that are relevant to MSM who won't have been vaccinated in school which could be drawn out if the emphasis of the paper were shifted a bit.

However, I also have some concerns about the methodology and the generalisability of the findings. As the authors note, the sample sizes are small, which I think is a particular issue for the survey. In addition, the response rate for the survey is (I presume?) impossible to calculate because of the recruitment methods. The authors acknowledge that the sample is self-selected and may differ from the wider population of MSM in significant ways, and I think this makes it a bit unclear how relevant the findings are beyond this very specific sample.

A few more specific points to consider when re-drafting the paper:

- I found the methods a little hard to follow at times, especially when it came to recruitment, and I wonder whether it would be clearer if the two sub-studies were described separately. For example, at the bottom of page 6, there is reference to combining the questionnaire responses from the focus group and online participants, but according to the sample size, this doesn't seem to have been done (unless some participants are double-counted in Table 1 if they took part in both studies?). I guess it's just important to understand if the 17 focus group participants were a sub-set of the 51 men who did the survey, or a separate sample.

- The Results are generally reported clearly but I wondered how missing data were treated? There seem to be missing data for all variables, and I wondered whether any participants should have been excluded because they didn't really answer any of the questions?

- The Discussion (p17) refers to comparisons between England and Northern Ireland but this comparison is not presented in the Results.

- Where comments are made about sample size, I'm not sure how the qualitative study can be deemed to have an 'appropriate' sample size if data saturation was not reached. I appreciate the challenges of recruiting hard-to-reach groups but making a pragmatic decision to stop recruitment doesn't necessarily mean the sample is large enough or that new themes and views wouldn't have been identified had there been resources to continue. For the survey study, a reference is made to 'power' but since no statistical analyses were carried out, I'm not sure what the study is underpowered to do.

REVIEWER	Roger Detels UCLA United States
REVIEW RETURNED	24-Jan-2019

GENERAL COMMENTS	This paper had a limited sample size and it is difficult to determine representativeness of respondents to an online survey. Clearly respondents differ from non-respondents. Value of paper is that it clearly indicates that targeting MSM and possible or undecided MSM at the recommended age, 12-13 years, is not a strategy that is feasible if you wish to reach MSM before their sexual debut. Clearly young men whether hetero or homosexual have a high likelihood of being infected and thus are at risk for penile cancer etc and if infected transmitting to their partners whether male or female. Low priority could be submitted as a brief report or letter.
---

VERSION 1 – AUTHOR RESPONSE

Reviewer(s)' Comments to Author:

Reviewer: 1

Reviewer Name: Holly Fontenot

Institution and Country: Boston College, USA

Please state any competing interests or state 'None declared': None declared

Please leave your comments for the authors below

Review:

Overall, very important topic and timely for the UK to consider for policy change to be more inclusive. Below are suggested revisions to add to the quality of the paper.

Thank you for your positive feedback.

Methods:

Pg 6 line 52- online via what platform, qualtrics?, secure? And how were advertisements done? Posters, ads on the organizations facebook pages or websites? Please describe how you did ads for the quant and qual parts of the study

We have provided more detailed information describing the survey and advertisements for both study components under Study Design in the section 'Questionnaire study' (page 7). We now write, "The survey was administered online using Survey Monkey and advertised via various Lesbian Gay Bisexual Transgender Queer (LGBTQ) organisations on social media (Twitter and Facebook).

Page 7 line 5- race and ethnicity. Also... this data was not presented in any table or text?

Thank you for spotting this omission. Ethnicity data is now reported in Table 1.

In methods ensure consistency in language for where you are recruiting. You say LGBTQ organizations and then you say advocacy groups. I would keep with just organizations.

LGBT is defined on page 7 but you first use the abbreviation earlier in the section on page 6.

Thank you for pointing out these inconsistencies. We have ensured consistency of the language by replacing “advocacy groups” with “organisations”. The abbreviation is now defined on page 7.

You describe how you recruited for the focus groups, but you did not describe the ads for how you recruited for the online survey.

Advertisements for the online questionnaire were placed on social media by various LGBTQ organisations. Please see page 9.

Data Collection:

Page 8, line 3-11. Did you also discuss effectiveness and recommendation to vaccinate even if the participant has past sexual experience, “catch up vaccination”- did they voice desire to go get vaccinated now?

Focus group participants were not asked to discuss their own sexual experiences and catch-up vaccination was not explicitly discussed beyond asking if anyone had been offered or requested the HPV vaccine. However, the participants expressed willingness to receive the vaccine if it were offered to them both now and when 12-13 years. This is described on page 13, “most participants were willing to receive the vaccine and wanted more information.”

I suggest a subheading to describe measures, then a separate heading to discuss data collection

Thank you for this suggestion. We have inserted separate headings describing measures and data collection as recommended. In response to reviewer 2, we have also divided the methods into a description of the questionnaire and focus group sub-studies.

Authors never really explained how quant data was recruited for or collected or where data was downloaded, managed, stored for analysis.

Recruitment is now described in more detail on page 9.

The data was downloaded from Survey Monkey into Excel and then uploaded to SPSS v12 for analysis (this information is now included on page 10).

Where the focus groups first then they informed the quant survey, did the focus groups and the public help to inform the survey? Make that clear. The section about the public is very confusing and not needed if that information is incorporated into the measures section of the paper.

The questionnaire and focus group topic guide were developed at the same time. The survey was informed by the Patient and Public Involvement work. All focus group participants were asked to complete the questionnaire before the focus group. Therefore, the focus groups did not inform the questionnaire. We have re-ordered the methods section for clarity and incorporated the Patient and Public Involvement section into the methods describing the survey development.

Analysis:

Break into 2 sections, 1) quant, 2) qual. For the quantitative it was all descriptive analyses, what analyses did you use, what software did you use?

The analysis section has now been divided into quantitative and qualitative sections. SPSS v12 was used to analyse the quantitative data. Frequencies and proportions were reported for categorical data and mean and standard deviation for continuous data. This information is reported on page 10.

Page 8 line 50. Please describe how you defined codes, then used that framework to move the subsequent coding forward, codebook? So beyond just comparing the 1st level codes, you build

consensus, then what was the plan, you must have agreed on definitions for codes to continue to move forward.

We have added the following on page 11 to clarify how the codes were defined and that consensus among the research team was sought at each stage of the analysis process:

“JK and CF independently coded the first transcript systematically, line-by-line, compared their coding and reached consensus on the definition of codes. These initial codes, which captured features of interest in the data, were then applied to the remaining transcripts. The content of all the codes was read and compared to each other to iteratively refine and cluster codes into themes and sub-themes. For example, duplicate codes with synonymous meanings were then collapsed. A description of each theme capturing instances of divergence was then written by JK. At each stage, findings were verified and discussed by the research team to assess accuracy and credibility of the interpretation, promote inter-rater reliability and ensure rigour.”

Page 9, line 10. Any remuneration for the participants? Survey or focus groups- spell that out here so when you mention that in the discussion it makes sense.

We have stated the following on page 11: “Participants were not provided any financial remuneration for their time.”

Results:

You can just say approximately 45 minutes, no need to give mean or range for how long the focus groups lasted.

This change has been made. Please see page 12.

Page 11. Tell me the 2 themes in line 6 in a sentence and then go into them in the subheadings

The following has been inserted on page 13: “Two main themes and several subthemes were elicited from the thematic analysis: 1) Willingness to be vaccinated and; 2) Implementation recommendations.”

Page 12. Can you give a sentence here again laying out the sub-themes before you go into them.

More richer description of participant discussion would be helpful, then use the quotes of exemplars for the findings.

The following has been inserted describing the sub-themes on page 14:

“Across the focus groups, recommendations to support the implementation of the HPV vaccine were gathered and grouped into two subthemes: ‘Promoting and raising awareness of the vaccine’ and ‘Identifying and offering YMSM the HPV vaccination’.”

Due to the journal word count constraints, the additional information added in response to comments on the methods and in the interest of keeping the manuscript concise a richer description of participant discussion has not been added.

Discussion:

Consider viewing these 2 articles: (1) Apaydin, Vaccine, 2018, DOI: 10.1016/j.vaccine.2018.06.057 and (2) Fontenot, Vaccine, 2016, 34, pg 6209-6216

The way in which a HCP recommends HPV vaccine to sexual and gender minority youth (affirming care) is important for vaccine uptake.

Thank you for highlighting these relevant articles. We have now referenced them within the discussion section.

Page 17. Line 26. “the sample size is however, considered appropriate as it has obtained a diverse range of views of YMSM”—how? What is the race, ethnicity- urban/ rural, educated, uneducated, homeless? Might be an overstatement

We agree that this statement could be viewed as an overstatement. The sentence has been deleted and replaced with the following: “Therefore, the findings must be read with caution.”

Participant ethnicity is now reported in Table 1.

Strengths and limitations could be re-worked.

End of that paragraph... does the UK have a vaccine “catch up” recommendation for those older than 12?

The UK does not currently have a catch-up programme for boys older than 12. We have commented on this in the Introduction and Discussion sections (page 6-7; page 19).

Tables: Consider a table with sample characteristics for the entire group with break outs for quant and qual groupings. Give general demographics and add race/ethnicity.

Table 1 presents the characteristics of the questionnaire and focus group participants.

Participant ethnicity is now reported in Table 1.

Reviewer: 2

Reviewer Name: Jo Waller

Institution and Country: UCL, UK

Please state any competing interests or state ‘None declared’: None declared

Please leave your comments for the authors below

I’m afraid the relevance of this paper has decreased significantly since it was submitted in June 2018, following the recommendation by the JCVI that gender neutral HPV vaccination be introduced in the UK. I’m not sure why there’s been such a long delay with reviewing the paper, which I only received in January 2019, and I sympathise with the inevitable frustration that the authors must feel about this.

Thank you, we appreciate your sympathy and the opportunity to revise the paper considering the JCVI recommendations and UK government decision to introduce gender neutral HPV vaccination. We still believe this research is helpful for: 1) understanding current YMSM who will not be offered the HPV vaccination in a gender neutral way (given there is no catch-up programme), and 2) in guiding other programmes internationally that do not have a gender neutral programme and are considering implementation of a programme for YMSM.

If the authors are still keen to publish the paper, major edits will have to be made to the Introduction and Discussion to re-frame the paper in the context of HPV vaccination for boys being introduced from the next academic year. This may be possible.

The abstract, introduction, discussion and conclusion sections now acknowledge the decision to vaccinate boys from 2019/2020 and consider the findings within this context. We emphasise the need

for this research in relation to the absence of a catch-up vaccination programme for boys and the sub-optimal uptake of the pilot targeted vaccination programme within GUM clinics.

Overall, the study addresses what was an important issue of how to maximise engagement with HPV vaccination among MSM. Much of the discussion (e.g. about trying to ascertain sexual orientation at a very young age, before initiation of sexual activity) is now moot, but there may still be findings that are relevant to MSM who won't have been vaccinated in school which could be drawn out if the emphasis of the paper were shifted a bit.

However, I also have some concerns about the methodology and the generalisability of the findings. As the authors note, the sample sizes are small, which I think is a particular issue for the survey. In addition, the response rate for the survey is (I presume?) impossible to calculate because of the recruitment methods. The authors acknowledge that the sample is self-selected and may differ from the wider population of MSM in significant ways, and I think this makes it a bit unclear how relevant the findings are beyond this very specific sample.

We have inserted a citation and accompanying text highlighting that sample size in research with sexual and gender minorities is often small.

A few more specific points to consider when re-drafting the paper:

- I found the methods a little hard to follow at times, especially when it came to recruitment, and I wonder whether it would be clearer if the two sub-studies were described separately. For example, at the bottom of page 6, there is reference to combining the questionnaire responses from the focus group and online participants, but according to the sample size, this doesn't seem to have been done (unless some participants are double-counted in Table 1 if they took part in both studies?). I guess it's just important to understand if the 17 focus group participants were a sub-set of the 51 men who did the survey, or a separate sample.

The two sub-studies are now described separately.

We have also confirmed on page 11 and Table 1 that the focus group participants were a subset of the 51 survey respondents.

- The Results are generally reported clearly but I wondered how missing data were treated? There seem to be missing data for all variables, and I wondered whether any participants should have been excluded because they didn't really answer any of the questions?

If participants indicated they were not sexually active they were asked to skip the sexual contact questions and if they indicated that they had never heard of the HPV vaccine they did not complete the knowledge/attitude scores. This information has now been inserted on page 10 to explain the missing data.

- The Discussion (p17) refers to comparisons between England and Northern Ireland but this comparison is not presented in the Results.

Thank you for identifying this omission. We now also report that there were no differences in experiences across the research settings on page 13.

- Where comments are made about sample size, I'm not sure how the qualitative study can be deemed to have an 'appropriate' sample size if data saturation was not reached. I appreciate the challenges of recruiting hard-to-reach groups but making a pragmatic decision to stop recruitment doesn't necessarily mean the sample is large enough or that new themes and views wouldn't have been identified had there been resources to continue.

We agree that this statement could be viewed as an overstatement. The sentence has been deleted and replaced with the following: "Therefore, the findings must be read with caution."

For the survey study, a reference is made to 'power' but since no statistical analyses were carried out, I'm not sure what the study is underpowered to do.

This line has been edited to read (see page 20): "The small sample size for the quantitative data resulted in a lack of statistical power to analyse data using inferential statistics and should be considered in generalising beyond the study sample."

Reviewer: 3

Reviewer Name: Roger Detels

Institution and Country: UCLA
United States

Please state any competing interests or state 'None declared': none declared

Please leave your comments for the authors below

This paper had a limited sample size and it is difficult to determine representativeness of respondents to an online survey. Clearly respondents differ from non-respondents. Value of paper is that it clearly indicates that targeting MSM and possible or undecided MSM at the recommended age, 12-13 years, is not a strategy that is feasible if you wish to reach MSM before their sexual debut. Clearly young men whether hetero or homosexual have a high likelihood of being infected and thus are at risk for penile cancer etc and if infected transmitting to their partners whether male or female. Low priority could be submitted as a brief report or letter.

Thank you for your feedback. The limitations on the study sample size are acknowledged in the 'Strengths and Limitations' section.